# Long-Horizon Reliability of LLM Agents: Social Exposure, Personas, and Metacognitive Policy on a Delay-of-Gratification Survival Benchmark

## Abstract

Large language models (LLMs) are increasingly deployed as long-horizon, multi-turn agents that must reason, plan, utilize tools, and interact with peers. Yet, most evaluations lack auditable, multi-factorial experiments with time-resolved statistics that reveal how behavior unfolds under explicit constraints. Inspired by the Stanford marshmallow experiment, we introduce a compact, multi-agent microbenchmark that reframes the delay of gratification as a discrete-time survival task. ReAct-style agents operate at minute-level granularity with an internal "raise-a-question" tool subject to a per-step budget; we factorially manipulate social visibility (broadcast vs. isolated), persona prompts (hedonic drive; age), and metacognitive policy (mandatory vs. optional self-questioning). From complete step-level traces we estimate Kaplan–Meier (KM) survival and discrete-time hazards, enabling transparent inspection of social influence and tool-use dynamics. We extend the study to 8 model families (open- and closed-weight), totaling 84,540 trajectories across 512 cells, with $\approx100\%$ valid runs. Aggregate behavior exhibits a sharp early impulse (initial eat 0.062) followed by a long low-hazard tail; completion is 0.824, with median time-to-eat $\approx17$ and Restricted mean survival time (RMST) $\approx16.47$. In pooled hazards, mandatory self-questioning increases per-minute risk ($\beta \approx 0.093$; Odds Ratio (OR) $\approx1.10$), while persona factors strongly modulate hazard (vs. crave: like OR $\approx0.45$, neutral $\approx0.26$, none $\approx0.24$; vs. adult: child $\approx8.65$, senior $\approx5.60$). The broadcast vs. isolated main effect is near zero on average ($\beta \approx -0.009$; OR $\approx 0.99$), but we uncover three hazard-shape regimes (near-flat, early-spike, and bi-modal) that vary by model family and mediate when social exposure matters. Ablations that remove hedonic and/or age instructions flatten hazards and raise completion toward 1.0. We release code, prompts, logs, and analysis artifacts to facilitate replication and future work on causal social exposure, networked interaction, and other long-horizon agent tasks.

## 1 Introduction

Modern uses of large language models (LLMs) are inherently conversational and iterative, as users and agents co-construct tasks over multiple turns, revise goals, and recover from mistakes. Recent multi-turn evaluations show that single-turn prowess does not guarantee long-horizon reliability: agentic setups reveal gaps in reasoning and decision-making Liu et al. (2023), tool use and natural-language feedback help but interact idiosyncratically with training and instruction tuning Wang et al. (2024b), and performance can drop substantially when moving from single- to multi-turn interaction Laban et al. (2025). These observations motivate *auditable, constrained, multi-factorial* experiments that measure how agent behavior unfolds over time and in the presence of other agents.

Inspired by a classic Stanford study on delayed gratification (Mischel & Ebbesen, 1972) and by recent LLM research replicating classical cognitive tasks (Lampinen et al., 2024; Strachan et al., 2024), including marshmallow-like scenarios (Coletta et al., 2024), we introduce cross-model relia-

bility study and present an auditable, minute-resolution benchmark that reframes delayed gratification as a discrete-time survival task for multi-turn LLM agents under explicit tool budgets.

## 1.1 CONTRIBUTIONS

**Benchmark.** We introduce a compact, auditable benchmark for long-horizon, multi-agent interaction. Episodes produce per-minute, per-agent traces (actions, messages, tool calls), enabling survival-analysis evaluation via KM curves, RMST, and discrete-time hazard models with time effects. This design yields transparent, step-level diagnostics suitable for ablations and policy comparisons.

**Cross-model generalization.** We evaluate eight LLM families and find that three core effects replicate, in direction and with similar order of magnitude, across models: (1) social exposure (broadcast vs. isolated), (2) metacognitive policy (MUST vs. MAY), and (3) persona (hedonic drive, age). We quantify between-family heterogeneity and pooled effects using per-family hazards and RMST with a random-effects meta-estimate and mixed-effects hazards.

**Ablations, dynamics, and policy.** Removing hedonic drive and/or age monotonically improves survival and narrows social-exposure gaps, while the MUST policy remains riskier than MAY across families. We characterize multi-turn dynamics with event-time distributions, step-wise hazards, tool-use trajectories, and peer-exposure patterns, yielding interpretable temporal profiles of failure that explain when social exposure harms most.

**Reproducibility and artifacts.** All experiments are fully scripted; we release per-minute logs, configuration files, prompts, and analysis code to regenerate every table and figure from raw traces (fixed seeds, hardware/cost details). We follow ICLR's reproducibility and ethics guidance, provide an anonymized artifact for review, and will open-source code and data upon acceptance.

## 1.2 HYPOTHESES AND FINDINGS

We test five hypotheses across eight models. **H1** *Social visibility*: when agents can observe peers, the hazard of committing to the immediate option increases relative to isolation. **H2** *Internal state*: personas reflecting stronger hedonic drive and child/senior age elevate hazard, whereas neutral drive and adult age reduce it. **H3** *Metacognition*: a mandatory self-questioning step (MUST) changes hazard relative to optional use (MAY); we assess whether such scaffolding stabilizes behavior. **H4** *Temporal structure*: the per-minute hazard is *non-constant* across the horizon (i.e., behavior displays systematic time dependence), without pre-specifying its shape. **H5** *Prompt crafting pre-registered expectation*: more prescriptive prompt scaffolding and instruction complexity, including enforced metacognitive steps, should improve adherence (lower hazard) compared to a minimalist design.

**Brief summary of findings (8 families, 84,540 trajectories):** **H1** Near-zero average social main effect: broadcast vs. isolated pooled OR $\approx 0.99$ (not significant). **H2** Strong persona effects: e.g., *child* and *senior* vs. *adult* OR $\approx 8.65$; *neutral* vs. *crave* OR $\approx 0.26$. **H3** MUST increases risk vs. MAY (OR $\approx 1.10$). **H4** Hazards are non-constant with an early spike (initial eat $= 0.062$). **H5** Contradicted: heavier prescriptive scaffolding does not lower hazard; MUST raises risk (H3), while removing persona prompts flattens hazards and pushes completion toward 1.0.

## 2 RELATED WORK

**Classic Cognitive Tasks, LLMs and Multi-Turn Interactions.** Our study contributes to a recent body of work that adapts classic cognitive tasks to investigate LLM capabilities. Models show human-like *content effects* in reasoning (Lampinen et al., 2024); near-human performance on Theory-of-Mind tasks can degrade under prompt variations (Strachan et al., 2024; Kosinski, 2024); and judgment, decision-making, and memory studies report framing/probability biases and capacity limits (Binz & Schulz, 2023; Wang et al., 2024a; Zhang et al., 2024; Gong & Zhang, 2024). These studies are largely single-prompt or short-horizon; we instead target *multi-turn* interaction by importing delayed gratification into a controlled, minute-by-minute, multi-agent setting.

**Human delay of gratification and intertemporal choice.** The Stanford marshmallow experiments and subsequent studies on delay of gratification show that attention and cognitive strategies modulate waiting, inspiring the "hot/cool" model of self-control (Mischel & Ebbesen, 1972; Metcalfe & Mischel, 1999). Long-term links to outcomes are moderated by environmental reliability and socioeconomic context (Kidd et al., 2013; Watts et al., 2018), with neural work implicating adult self-control circuitry (Casey et al., 2011). A recent study explored marshmallow-like scenarios in the context of LLMs (Coletta et al., 2024), although it did not include temporal/social analyses. In behavioral economics, intertemporal choice formalizes conflicts between immediate and delayed rewards via hyperbolic discounting and time-inconsistent preferences (where immediate rewards are disproportionately valued over future ones, formalized in the $\beta$-$\delta$ model), commitment, and naïve vs. sophisticated agents (Ainslie, 1992; Laibson, 1997; O'Donoghue & Rabin, 1999); classic procedures quantify delay preferences (Mazur, 1987). We adapt these insights to an LLM survival-analysis frame over discrete minutes.

**Scaffolding, multi-agent interaction, evaluation, and personas.** Our framework builds on several key developments in LLM interaction design and evaluation. Reasoning architectures such as Re-Act, Self-Ask, and Reflexion scaffold stepwise deliberation and tool use (Yao et al., 2023a; Press et al., 2022; Yao et al., 2023b; Shinn et al., 2023). Multi-agent coordination and social simulation, e.g., debate, role-based systems, and long-horizon agent societies, provide structure for interaction and influence (Du et al., 2023; Li et al., 2023; Wu et al., 2024; Park et al., 2023). Our analysis uses time-to-event tools, KM and discrete-time logistic hazard models, to quantify factor effects on waiting (Kaplan & Meier, 1958b; Singer & Willett, 1993; Allison, 1982). Finally, persona prompting distinguishes role-play from personalization (Tseng et al., 2024); although personas may not improve objective task performance and can bias behavior (Zheng et al., 2024), we employ them as controlled manipulations while acknowledging the limitations of LLMs as human surrogates (Gao et al., 2025)

## 3 EXPERIMENT SETTING

**Environment and Episode Modeling.** We evaluate LLM agents in a finite-horizon, multi-turn environment formalized as a Partially Observable Markov Decision Process (POMDP) Figure 1a (Kaelbling et al., 1998), characterized by horizon $H = risk\_horizon + 1$. At each step $t \in \{0, \ldots, risk\_horizon\}$, the environment state $S_t$ evolves based on the agent's action $A_t$. The agent receives an observation $O_t$, composed of the current step index and, in broadcast conditions, recent peer actions ($\texttt{others\_responses}_t$). After optional internal deliberation via the $\texttt{raise\_a\_question}$ tool (limited by a per-step budget), the agent emits a constrained action $A_t \in \{\text{I wait}, \text{I eat the marshmallow}\}$. The environment returns a reward $R_{t+1}$, increments the step $t \to t+1$, and provides the next observation $O_{t+1}$. Agents reaching the end of the risk horizon without eating move to the threshold step ($threshold\_step = risk\_horizon + 1$), receiving the delayed payoff. Formally, the interaction loop at each step is:

$$O_t = [\text{Time}(t), \ \mathbb{1}_{\text{broadcast}} \cdot \texttt{others\_responses}_t],$$

$$A_t = \pi_\theta \left(O_t, \ \{\texttt{raise\_a\_question}(O_t, i)\}_{i=1}^{k_t}\right), \quad 0 \le k_t \le cap,$$

$$R_{t+1}, S_{t+1}, O_{t+1} = \mathcal{E}(S_t, A_t),$$

$$b_t(S_t) = P(S_t \mid O_{0:t}, A_{0:t-1}),$$

where the tool is *internal* and does not alter $S_t$, and in isolated conditions ($\mathbb{1}_{\text{broadcast}} = 0$), observations $O_t$ fully determine the underlying state $S_t$, reducing the environment to a finite-horizon Markov Decision Process (MDP) Figure 1a (Puterman, 1994)).

The agents operate within a ReAct loop (Yao et al., 2023a) (Thought + Tool $\to$ PAUSE $\to$ Observation $\to$ Thought + Answer) with a dedicated validation tool, $\texttt{raise\_a\_question}$, which is gated by a per-step budget. The environment is designed to be turn-based and synchronous: at each minute, all active agents observe, decide, and act. When an agent chooses to eat, they are eliminated from subsequent minutes, while waiting maintains the agent's participation but keeps them at risk.

We implement a factorial design that manipulates agents' social context (isolated vs. broadcast), internal personas (hedonic drive and age), and metacognitive scaffolding (mandatory vs. optional tool use). We assess these factors via KM curves (Kaplan & Meier, 1958a) and discrete-time hazard models (Allison, 1982).

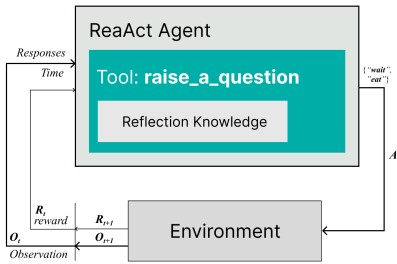
(a) Episode Modeling as POMDP.

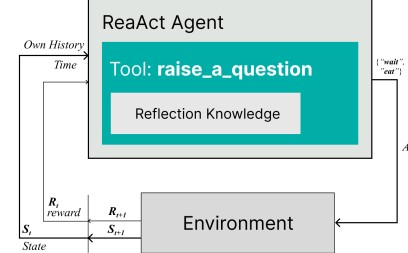
(b) Episode Modeling as MDP.

Figure 1: Interaction loops for (a) the partially observable Markov decision process (POMDP, broadcast condition) and (b) the fully observable Markov decision process (MDP, isolated condition). In (a), the agent observes time and peer responses, introducing partial observability. In (b), the agent observes only time and its own history, rendering the environment fully observable. At each step, the agent internally uses a capped `raise_a_question` tool, then chooses an action $a_t \in \{\text{wait}, \text{eat}\}$. The environment provides a reward, updates the state, and advances to the next observation.

**Time Horizon & Rewards.** The scenario maps 20 minutes of delayed gratification to $T$ discrete steps (default $T = 20$, with Step 0 serving as initialization). As LLMs do not have an inherent concept of time, the time is modeled in natural language: at each step, the agent is reminded that $Xth$ minutes has passed. The environment implements a reward structure where agents receive +1.0 for outputting "I eat the marshmallow" at any minute $t$, followed by elimination. Agents who persist until the final minute receive a terminal reward of +2.0, representing successful delayed gratification. For implementation purposes, the final step employs a final-resolution prompt. ReAct agents are forced to return {Answer: "I won"} to signify the successful resolution of the session. This response is normalized to "waited_full" during analysis while preserving raw traces.

**Social Context & Personas.** We manipulate the observability of peers through two distinct conditions: *Isolated:* Agents have access only to their own historical actions and outcomes; *Broadcast:* Observations include structured summaries of peers' last actions history per step and other responses. This design enables the study of social influence pathways, such as cascading effects when peers opt for early consumption. Each agent is parameterized by persona prompts that incorporate reflection knowledge, such as age and hedonic drive. The age persona can be *child (5 y/o), adult (35 y/o), senior (75 y/o), none("You are a policy")*, while the hedonic drive can be categorized as *crave, like, neutral, none*. These instructions remain private to the agent and are explicitly referenced in its Thought traces (e.g., "I am a 75 years old and I crave sweets.").

**Tool Use & Metacognition.** ReAct Agents employ *raise_a_question* tool for self-querying under a per-step budget constraint. We vary two key aspects: *tool_policy:* agents either MUST use the tool or MAY use it optionally. Agents are unaware of their available question budget and are directed to provide and answer if the budget is exhausted. Budget-related metrics are logged at each step to facilitate downstream analyses.

**Action Space Integrity & Validation.** The environment implements strict validation of terminal answer formats at each minute. Only the exact strings "I eat the marshmallow" or "I wait" are accepted as valid responses. Steps containing any other response are marked with a validation error in the trajectory. Agents who choose to eat are recorded along with their termination metadata.

**Implementation & Reproducibility.** The framework maintains a clear separation between abstractions and scenario plugins, providing a single-entry run and evaluation harness for parameter sweeps. Each experimental run captures a full set of agents' trajectories which includes: a full ReAct traces (Thought, Tool calls, Observations, Answers); budget accounting episode summaries consolidating per-agent outcomes, timing, and rewards; enhanced analytics covering per-step social exposure in broadcast mode, tool usage patterns, and data-quality signals. Downstream processing scripts generate analysis-ready CSVs and reports featuring KM curves, hazard plots, and model results. The source code for our experimental framework and analysis will be made publicly available on GitHub upon publication; a direct link is withheld to preserve the integrity of the double-blind review process. The experimental setup configuration and agent prompts are provided in the Ap-

pendix. Supplementary materials include aggregated data for each model and analysis script, along with use instructions.

## 4    METHODS AND PROCEDURE

**Experimental Factors.** We implement a factorial design that crosses several core dimensions, with independent randomization per experimental cell and replicated trials: social context: isolated vs. broadcast; hedonic drive: crave vs. like vs. neutral; age persona: child vs. adult vs. senior; tool-use policy: MUST vs. MAY. Optional toggles in the run plan include budget visibility (visible vs. hidden). The default total time is set to `max_steps = 20` (minutes), with a fixed answer format and final reward of +2.0.

**Agents & Reasoning Loop.** All agents are LLM-driven using **Gemini-2.5-Flash** (same model across all cells and trials). The ReAct agent executes the following loop for each minute $t$ in $0..T$:

---
**Algorithm 1** Agent Reasoning Loop per Minute

---
1: **Environment:** Observation: Timestamp (minutes passed) and History of Responses (Own or All)
2: **Thought:** reflect given persona & current observation
3: **Tool** (optional or required): `raise_a_question` ($\leq$ per-step budget)
4: **PAUSE**
5: **Observation:** tool return, plus environment update (incl. peers if broadcast)
6: **Thought:** integrate tool feedback & social signals
7: **Answer:** exactly "I eat the marshmallow" or "I wait"

---

Budget enforcement ensures that exceeding the per-step cap forces the response and prevents further tool calls within that minute.

**Procedure:** The experimental procedure consists of three phases: *(1) Initialization (Step 0):* Agents receive the starting prompt and are expected to make their first decisions; *(2) Main loop (Minutes 1..T):* At each minute, the environment processes last actions, issues rewards for eaters, updates observations (including social stats), and requests next actions from active agents. *(3) Final minute:* The environment issues a final-resolution prompt; remaining agents commit to waiting and receive +2.0. Any internal {Answer: "I won"} is registered as "waited full".

**Data & Logging.** For each agent × step interaction, we log: *decision & reward* (action, reward, termination flags); *tool usage; validation*(format compliance and error counts); *social exposure*(peers waiting, peers eliminated, eats per step, waits per step); *run metadata* (model settings, temperature, seed, scenario parameters). The pipeline compiles agent outcomes, step-level trajectories, cell aggregates, and cell summaries.

### 4.1    METRICS AND STATISTICAL ANALYSIS

We model time-to-give-in as a discrete-time survival process. The event is the first minute an agent outputs "I eat the marshmallow"; agents who never eat by the horizon $T$ are right-censored at $T$ and coded as "waited_full." Invalid steps (format violations) are tracked and excluded per pre-specified rules.

**Restricted mean survival time (RMST).** As a scale-interpretable summary, we report RMST (Irwin, 1949; Royston & Parmar, 2013) up to $\tau$ minutes, i.e., the area under the survival curve truncated at $\tau$. In our minute-level design,

$$\widehat{\mathrm{RMST}}(\tau) = \sum_{m=0}^{\tau-1} \widehat{S}(m), \qquad \widehat{S}(0) = 1,$$

Table 1: Dataset overview (8 model families): counts, survival metrics, and data quality.

| Data Summary | | Survival Metrics (overall) | | Data Quality | |
|---|---|---|---|---|---|
| Model Families | 8 | Initial Eat Rate | 0.062 | Valid Rate | 99.98% |
| Total Agent Trajectories | 84,540 | Total Eat Rate | 0.176 | Valid Trajectories | 84,525 |
| Total Cells | 512 | Winners Rate | 0.824 | Invalid Trajectories | 15 |
| Risk Horizon ($T$) | 19 | Median TTE (steps) | 17.0 | | |
| | | RMST (steps) | 16.47 | | |

where $\widehat{S}(m)$ is the KM survival estimate at the start of minute $m+1$. We compute condition-wise RMST (and differences where noted) with 95% CIs from a nonparametric bootstrap (clustered by trial).

**Kaplan-Meier (KM) Survival Curves.** We employ KM survival curves (Kaplan & Meier, 1958b) to estimate and visualize the survival function. In this context, "survival" refers to an agent continuing to wait for the larger reward. The analysis plots the probability of an agent not having "eaten the marshmallow" at each discrete minute of the experiment. Survival probabilities are calculated at each step, KM plots are generated for each experimental factor (e.g., communication mode, hedonic drive). To represent uncertainty in the estimates, 95% confidence intervals are calculated using the Greenwood formula (Kaplan & Meier, 1958b; Greenwood, 1926; Klein & Moeschberger, 2003).

**Discrete-Time Hazard Model.** To quantify the effect of experimental factors on agent decisions, we use a discrete-time hazard model. This analysis estimates the impact of each factor on the probability of an agent "eating the marshmallow" at a specific time $t$, given they have survived (i.e., waited) until that point. This conditional probability is the hazard rate.

The analysis is implemented using a logistic regression model, a type of Generalized Linear Model (GLM), on the granular agent-step-level data. The detailed model specification is provided in the Appendix.

**Social-Influence and Tool-Use Dynamics:** While the hazard model focuses on the effects of time-invariant experimental conditions, we also analyze the dynamics of social influence and tool use through detailed visualizations illustrating the average number of peers observed eating or waiting at each step, providing insight into the social signals agents receive; and average number of "questions asked" (tool uses) by agents at each step, indicating metacognitive activity. This descriptive analysis of how social signals and metacognitive actions unfold over time complements the inferential hazard model.

## 5 RESULTS

**Sample and Data Quality.** We ran 84,540 agent trajectories spanning 8 model families and 512 experimental cells (time horizon $T = 19$). Data quality was near-perfect: 84,525 valid runs (99.98%) and 15 invalid (0.02%). Aggregate behavior shows a strong first-minute impulse followed by a long low-hazard tail. Key metrics: initial eat rate = 0.062, total eat rate = 0.176, and winners rate = 0.824. The median time-to-eat was $\approx 17$ minutes, with a restricted mean survival time (RMST) of $\approx 16.47$. Table 1 summarizes counts and validity, and Figure 2 visualizes the survival profile for each of the eight models.

### 5.1 MAIN EFFECTS

**Social context shifts risk.** Pooled across 8 model families and 512 cells, the broadcast vs. isolated *main effect* on per-minute hazard is near zero ($\beta \approx -0.009$, OR $\approx 0.99$, not significant). Descriptively, completion is marginally higher in broadcast than isolated (winners 0.827 vs. 0.822; total-eat 0.173 vs. 0.178), and median time-to-eat is nearly identical in both conditions ($\approx 17$ steps). However, families exhibit distinct *hazard-shape regimes*—near-flat (e.g., GPT-4o), early-spike (e.g., Gemini, Qwen, GPT-OSS-20B, DeepSeek-3.1), and bi-modal (e.g., Llama-3.1-8B,

DevStrall-Small-2505), which explains why the pooled broadcast effect averages to $\approx 0$; social exposure matters *when* it appears (early in left-spike families; late in bi-modal families). Figures are provide in Appendix 12.

**Internal drives and Age personas.** Persona factors strongly modulate hazard. Relative to *crave*, *like* reduces risk (OR $\approx 0.45$), as do *neutral* (OR $\approx 0.26$) and *none* (OR $\approx 0.24$). Relative to *adult*, *child* greatly increases risk (OR $\approx 8.65$) and *senior* also elevates risk (OR $\approx 5.60$). Removing hedonic and/or age instructions flattens hazards and raises completion toward $1.0$.

**Metacognition (tool policy).** Enforcing self-questioning increases risk: MUST vs. MAY yields $\beta \approx 0.093$ (OR $\approx 1.10$, $p \ll 0.001$). Tool-use telemetry shows higher question-tool calls under MUST, coinciding with earlier commitment events rather than stabilizing behavior.

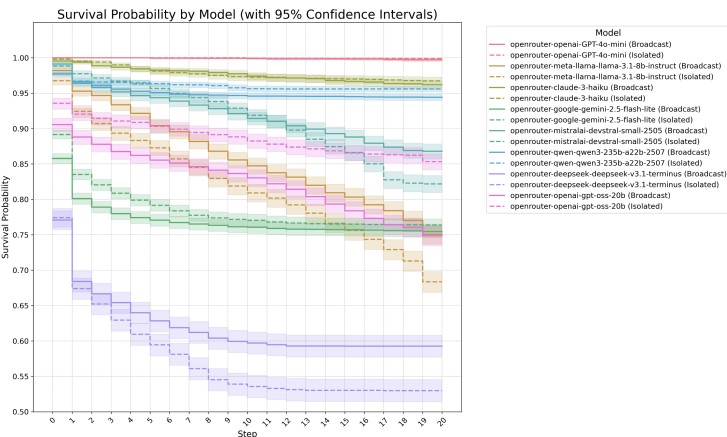

Figure 2: Kaplan-Meier survival (waiting) curves for 8 models.

## 5.2 INTERACTION DYNAMICS

**Reasoning dynamics under social exposure.** Across 8 model families and 84,540 trajectories, question-asking (our proxy for deliberation) declines over time in both social conditions. When pooled, the broadcast and isolated curves are nearly overlapping, and the average difference in per-step question rate is small, consistent with the near-zero broadcast main effect on hazard reported above. However, the pattern is regime-dependent: in *early-spike* families, questioning drops sharply in the first minutes under broadcast; in *bi-modal* families, we observe a late uptick in questioning near steps 16-19; and *near-flat* families maintain a low, steady rate throughout. Step-level peer exposure (number of peers who ate at $t-1$) co-varies with these phases and predicts higher hazard at the corresponding times, indicating time-varying social contagion rather than a uniform average effect. Figure 3 plots mean questions per step by social condition, and Figure 4 shows the peer-exposure traces that align with the early/late phases in the hazard-shape regimes.

## 5.3 ABLATIONS

We conducted targeted ablations to identify the source of multi-turn failures. First, we set *hedonic* to *none*; second, we removed the *persona age* (set to *none*); third, we removed *both* simultaneously. Each ablation was crossed with social context (broadcast vs. isolated) and tool policy (MUST vs. MAY).

**High-level results. Persona removals flatten risk and raise completion.** Across 8 model families, removing hedonic and/or age prompts consistently reduces early hazard and increases survival relative to the full-persona baseline (pooled winners $= 0.824$). The *combined* ablation (no hedonic, no age) yields the largest improvement, pushing completion toward $1.0$ and compressing the broadcast–isolated gap; single-factor ablations (*no hedonic* or *no age*) show intermediate gains. Hazard-shape diagnostics show the early spike is strongly attenuated under ablations, with survival

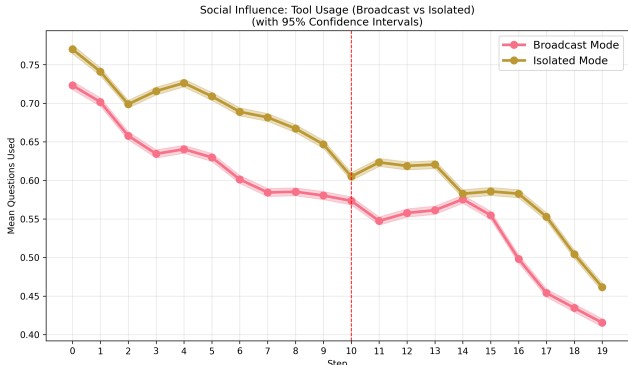

Figure 3: Mean questions per step with 95% CIs, split by social visibility (broadcast vs. isolated). Rates decline over time in both conditions and are close when pooled, consistent with the near-zero broadcast main effect on hazard.

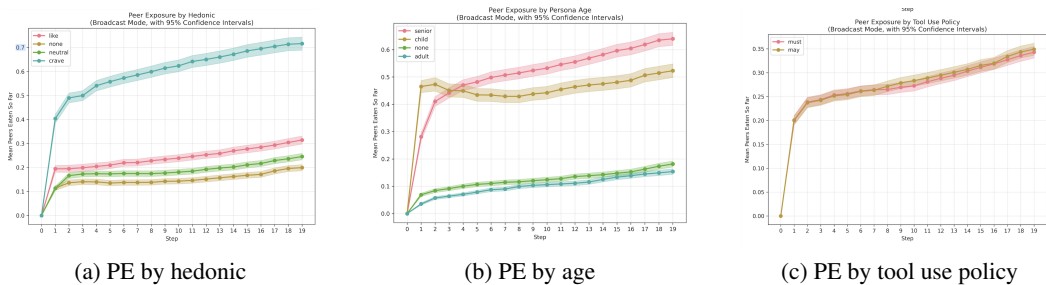

| (a) PE by hedonic | (b) PE by age | (c) PE by tool use policy |

Figure 4: Peer exposure(PE) (fraction of peers who have eaten so far) over time with 95% CIs, stratified by hedonic persona, age persona, and tool policy. Early exposure is highest under *crave* and *child*, aligning with early-spike hazard regimes; exposure grows later in runs for other settings.

approaching a near-flat profile through the horizon. Figure 9 summarizes completion by ablation condition; detailed survival shapes are provided in the appendix.

## 6 DISCUSSION

Our results reveal that simple semantic manipulations significantly impact failure rates in multi-turn interactions. Three key insights emerge. First, we confirm that the consistent pattern of early temptation followed by low-hazard persistence validates this framework as a stress test for long-horizon reliability. Second, persona-based internal states (hedonic drive, age) systematically affect survival, offering controlled probes of long-horizon stability. Third, the use of mandatory tools increases the hazard, suggesting that front-loaded deliberation may focus attention on temptation at critical decision points. The survival-signature and social-exposure effect appear model-agnostic across our evaluated families, whereas the impact of mandatory self-questioning varies by family, highlighting heterogeneity in reasoning scaffolds.

**Limitations.** We introduced a novel marshmallow-inspired, multi-agent Benchmark that turns delayed gratification into a tractable, auditable test of multi-turn reliability in LLM agents, and we hope this framework will serve the community as a compact testbed for studying self-control, social spillovers, and tool-use policies in multi-turn, multi-agent LLM systems. However, we acknowledge that our experiment has several limitations. First, our experiments uses a fixed decoding setup; we did not sweep temperatures or other sampling parameters, so cross-model/decoder generalization remains unknown. Second, the task is a single micro-environment with a strictly binary action space ("I eat the marshmallow" vs. "I wait") and a fixed reward scheme, which simplifies real deployments. Third, social context was varied only between the extremes of *isolated* and *broadcast*; richer network structures or partial observability were not explored. Next, question-budget visibility was

held *hidden* in the reported runs (no variation), so we cannot isolate awareness effects. Finally, our discrete-time hazard model includes time dummies and condition indicators, but omits time-varying peer-exposure and tool-use covariates (analyzed descriptively), which limits causal claims about social cascades.

**Implications and Future Work.** These findings have implications for multi-turn agentic systems that require sustained adherence, including carefully managing social exposure when cascading failures are possible, preferring optional over mandatory tool policies, and monitoring step-level metrics to detect early impulses. Our controlled setup (strict action space, scripted personas)provides a reproducible testbed for studying multi-turn reliability. In future work, we will explore other models, reward structures, and ways to generalize our approach to open-ended tasks.

## 7 CONCLUSION

We presented a marshmallow-inspired, long-horizon micro-benchmark that evaluates multi-turn LLM agents under controlled social contexts, personas, and tool-policy manipulations. Formalized as an MDP (isolated) and a POMDP (broadcast), the environment yields auditable, time-resolved traces that we analyze using KM survival and discrete-time hazard models. Empirically, broadcast peer visibility increases early-eat hazard, mandatory self-questioning raises risk, and persona factors (hedonic drive, age) strongly modulate waiting behavior. Together, these results demonstrate that social exposure and metacognitive scaffolding significantly influence temporal decisions in LLM agents. In relation to our hypotheses, the evidence indicates that social visibility elevates risk while isolation reduces it (H1), internal state manipulations systematically shift hazard (H2), mandatory metacognition increases rather than lowers risk (H3), the decision process has clear time dependence with an early spike and long tail (H4), and, contrary to expectation, more prescriptive prompt scaffolding does not improve adherence and can degrade reliability (H5). Future work will test broader model families, randomized social schedules for causal leverage, and additional tasks that stress tool budgets and coordination beyond delay of gratification.

## 8 REPRODUCIBILITY

We provide an anonymized supplementary zip archive containing aggregated data, analysis scripts, sample prompts, trajectories, and configurations to regenerate all figures and tables from the collected data. The main paper specifies the task formalization and failure/event definitions; the appendix details the contents of the supplementary materials, including implementation choices (environment, prompts, and models), as well as the complete statistical pipeline, encompassing Kaplan–Meier estimation, discrete-time hazard modeling, and significance testing procedures. We also release the experiment configuration. Analysis scripts enable end-to-end regeneration of results from configuration to plots.

## 9 ETHICS STATETMENT

All authors have read and will adhere to the ICLR Code of Ethics [1] . Our study evaluates synthetic interactions among large language models in controlled environments; it involves no human participants or personally identifiable data, and therefore did not require the Ethics Board review at our institution. Persona prompts and social-exposure conditions are used solely as experimental stylizations of model behavior; we do not target or stereotype real demographic groups. All third-party models and APIs were used in accordance with their terms and licenses. We are unaware of conflicts of interest that could bias this work; any that arise will be transparently reported in the camera-ready. As non-native speakers we used LLMs to polish the writing. LLMs were also used during code packaging.

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

## A  APPENDIX

### A.1  REPRODUCIBILITY PACK

We provide an anonymous `reproduce_analysis.zip` containing: (i) per–model aggregated CSVs (`agent_outcomes.csv`, `step_level_data.csv`, `cell_aggregates.csv`, `cell_summary.csv`), (ii) analysis scripts (`scripts/`, `analyze_standalone.py`) and one–shot runner (`reproduce_all.sh`), (iii) a short `README.md` with a one–command rebuild and figure mapping, (iv) prompts/configs (this appendix also reproduces key snippets verbatim), (v) a manifest of headline metrics and model outputs.

**One–command rebuild.**

```
bash reproduce_all.sh # regenerates report from included CSVs
```

**Anonymity and full code.**  Full environment/agent and data-aggregation code will be released upon acceptance; the review ZIP contains all aggregated data and analysis needed to verify every number and figure.

### A.2  DATASET SCALE, QUALITY, AND HEADLINE STATISTICS

Table 2 summarizes global statistics computed from the included CSVs.

Table 2: Overall summary (all 8 model families).

| | |
|---|---|
| Initial eat rate | 0.0616 |
| Winners rate | 0.8245 |
| Median time-to-eat (steps) | 16.9785 |
| RMST (steps) | 16.4716 |
| Valid outcomes (N) | 84525 |
| Invalid outcomes (N) | 15 |
| Data quality rate | 0.9999 |

Communication outcomes by social condition (mean across cells) are in Table 3.

Table 3: Broadcast vs. Isolated (means across cells).

| Condition | Winners rate | Median TTE | RMST |
|---|---|---|---|
| Broadcast | 0.8272 | 16.97 | 16.48 |
| Isolated | 0.8218 | 16.99 | 16.47 |

### A.3 STATISTICAL MODEL DETAILS (PRIMARY)

**Descrete time model hazard specification**:

Let $h_i(t)$ be the hazard for agent $i$ at time $t$. The model is specified as:

$$\text{logit}(h_i(t)) = \log\left(\frac{h_i(t)}{1 - h_i(t)}\right) = \alpha_t + \mathbf{X}_i^T \boldsymbol{\beta} \tag{1}$$

where $\alpha_t$ represents a set of time dummies that capture how the baseline probability of eating changes over time; $\mathbf{X}_i$ is a vector of covariates representing the experimental conditions for agent $i$ (e.g., communication mode is textitbroadcast, the hedonic drive level is *crave*, the persona age is *child*, etc.); $\boldsymbol{\beta}$ is the vector of coefficients that quantify the effect of each factor on the log-odds of eating. For instance, a positive coefficient for *broadcast* would imply that being in the broadcast condition increases the hazard of eating compared to the *isolated* condition.

We fit a discrete-time logistic hazard model with step indicators (0–19), cluster-robust SEs at the trial level, and factors: `communication_type` $\in$ {broadcast, isolated}, `tool_use_policy` $\in$ {may, must}, `hedonic` $\in$ {crave, like, neutral, none}, `persona_age` $\in$ {adult, child, senior, none}. We report coefficients ($\beta$), odds ratios (OR $= e^{\beta}$), and $p$-values. The pooled results are in Table 4.

Table 4: Pooled discrete-time hazard model (event-at-$t$).

| Contrast | $\beta$ | OR | $p$ |
|---|---|---|---|
| MUST (vs MAY) | 0.093 | 1.097 | 3.81e-11 |
| Isolated (vs Broadcast) | -0.009 | 0.991 | 5.14e-01 |
| Hedonic: like (vs crave) | -0.807 | 0.446 | $< 10^{-300}$ |
| Hedonic: neutral (vs crave) | -1.331 | 0.264 | 7.64e-123 |
| Hedonic: none (vs crave) | -1.439 | 0.237 | $< 10^{-300}$ |
| Age: child (vs adult) | 2.157 | 8.647 | $< 10^{-300}$ |
| Age: senior (vs adult) | 1.723 | 5.603 | $< 10^{-300}$ |
| Age: none (vs adult) | -0.102 | 0.903 | 2.24e-02 |

### A.4 PROMPTS AND EXPERIMENTAL MATERIALS (VERBATIM EXCERPTS)

ENVIRONMENT YAML TEMPLATE (EXCERPT)

```
max_steps: 20
question_budget: 3

rules_prompt:
  - You can choose to eat your single marshmallow at any time by saying
    "I eat the marshmallow".
  - If you eat the marshmallow your waiting is over.
  - However, if you can wait for {max_steps} minutes without eating it,
    you will receive 2 marshmallows.

answer_format: >
  CRITICAL RULE: In your JSON {Answer: <value>} you must only produce
  one of: "I eat the marshmallow" | "I wait".
```

REASONING/TOOL POLICY PROMPT (EXCERPT)

```
<loop_rules>
CRITICAL TOOL USE INSTRUCTION:
YOU {tool_use_policy} USE TOOLS TO VALIDATE YOUR DECISIONS
{visible_question_cap}
</loop_rules>
```

```
<tool_list>
raise_a_question ... You have a limited number of questions you can ask.
</tool_list>
```

### DESIGN FACTORS AND DECODING

Factors (full matrices and seeds in the ZIP):

- **Social:** communication_type $\in$ {broadcast, isolated}

- **Policy:** tool_use_policy $\in$ {may, must}

- **Persona (hedonic):** {crave, like, neutral, none}

- **Persona (age):** {child, adult, senior, none}

- **Visibility toggles:** visible_question_budget=false, distraction=none

- **Decoding:** temperature=0.5, top_p=0.7, max_tokens=1000

- **Global caps:** max_steps=20, question_cap_per_minute=3, reasoning_step_cap_total=10

## A.5 AGENT LOOP AND VALIDATION RULES

**Validation rules.** Only two canonical actions are valid at each minute: `"I eat the marshmallow"` or `"I wait"`. Non-canonical terminal strings that imply success (e.g., `"I won"`) are normalized to `waited_full` at $t=19$ unless an eat-event was recorded earlier.

**Loop pseudocode.**

```
for t in {0..19}:
  prompt = build_prompt(system, persona, policy, minute=t, social_context, history)
  reply  = model(prompt, temperature, top_p, max_tokens)
  action = normalize(reply)  # "I eat the marshmallow" | "I wait" | misc.
  if action == "I eat the marshmallow": record event at t; break
  elif action == "I wait": record censor at t; continue
  If no eat at t=19, mark waited_full
```

## A.6 EXAMPLE TRAJECTORIES (VERBATIM EXCERPTS)

Sample Full YAML/JSON trajectories are included in the ZIP; we show short, verbatim excerpts for orientation.

**Example A** (model_id=`GPT-4o-mini`, policy=MAY, persona=adult/neutral, comm=isolated):

```
t=0 user: "No time has passed yet...
    You are in a room alone"
t=0 assistant: Thought: ...
    Tool: raise_a_question: What would be the appropriate strategy?  PAUSE
t=1 user: "Minute 1 of 20 has passed... What will you do?"
t=1 assistant: Thought: ... Tool: raise_a_question: ...
...
t=17 assistant: Thought: ... {{Answer: "I eat the marshmallow"}}
```

**Example B** (model_id=`qwen3-235b`, policy=MUST, persona=child/crave, comm=broadcast):

```
t=0 user: "No time has passed yet... You are in a room together."
t=0 assistant: Thought: ... Tool: raise_a_question: ...
t=1..19 assistant: ... {{Answer: "I wait"}}
Final: normalized as waited_full at t=19
```

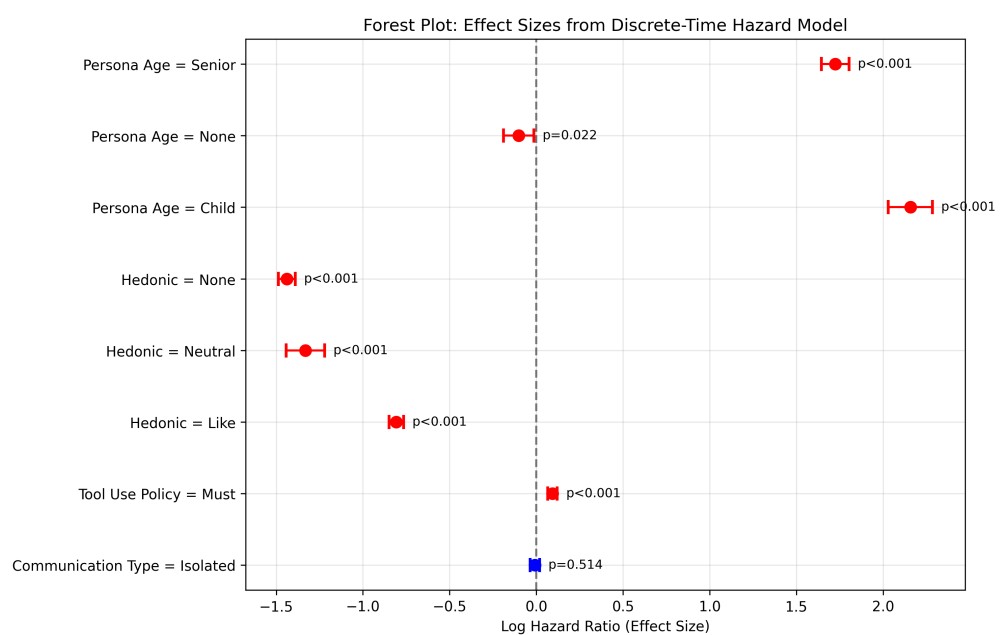

Figure 5: Effect-size forest plot (pooled ORs with CIs).

### A.7 FIGURES REPRODUCIBLE FROM `REPRODUCE_ANALYSIS.ZIP`

### A.8 ABLATIONS

**Additional diagnostics.** Figure 8 shows Kaplan-Meier survival under each ablation, confirming that persona removals suppress the early spike and yield flatter hazards throughout the horizon. Figure 11 provides a compact completion comparison (strict vs. relaxed policy view) consistent with the main text. Figure 10 reports question-tool dynamics: ablations lower per-step question rates, while the MUST policy maintains higher usage and corresponds to higher hazard, matching our pooled hazard estimates.

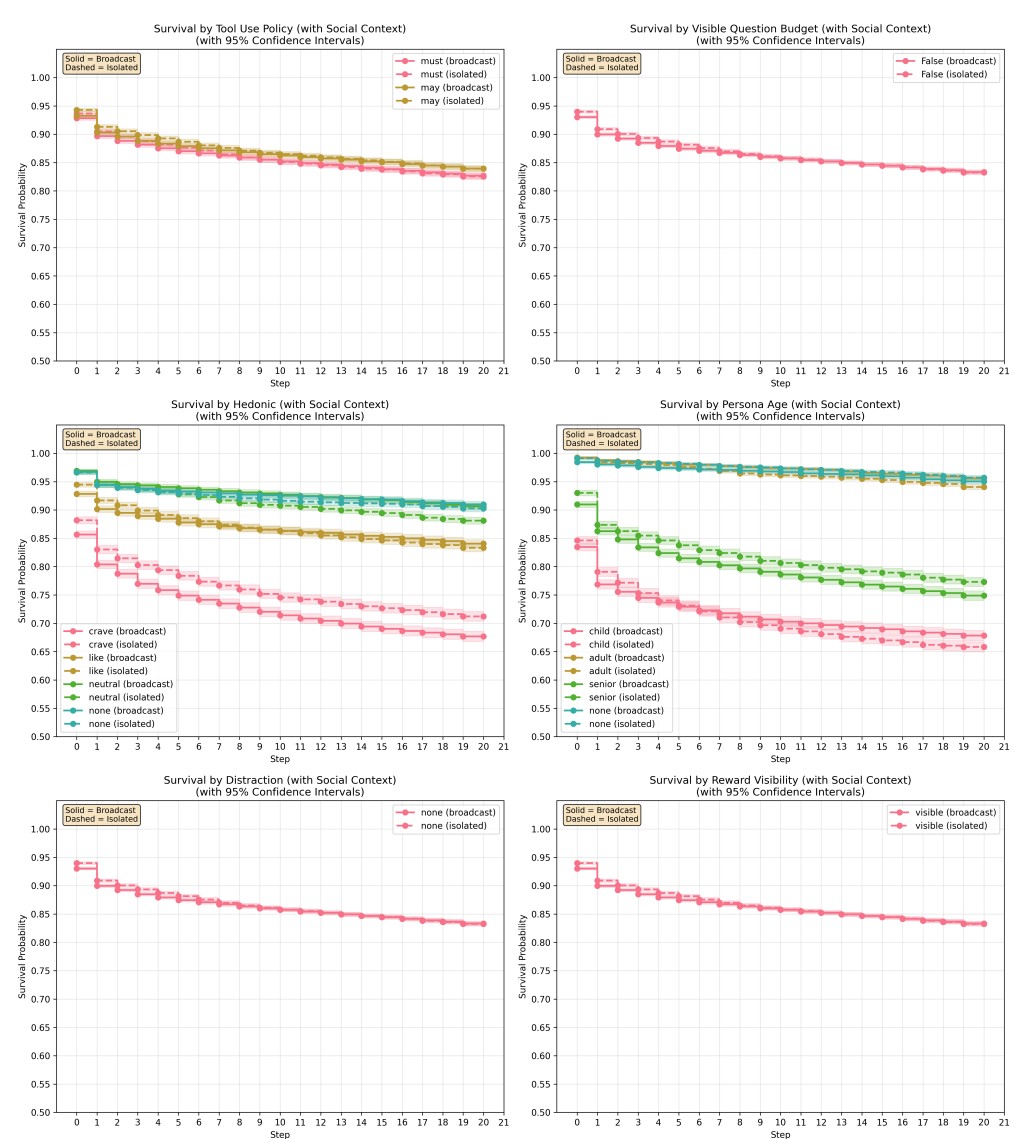

Figure 6: Kaplan–Meier survival across all families.

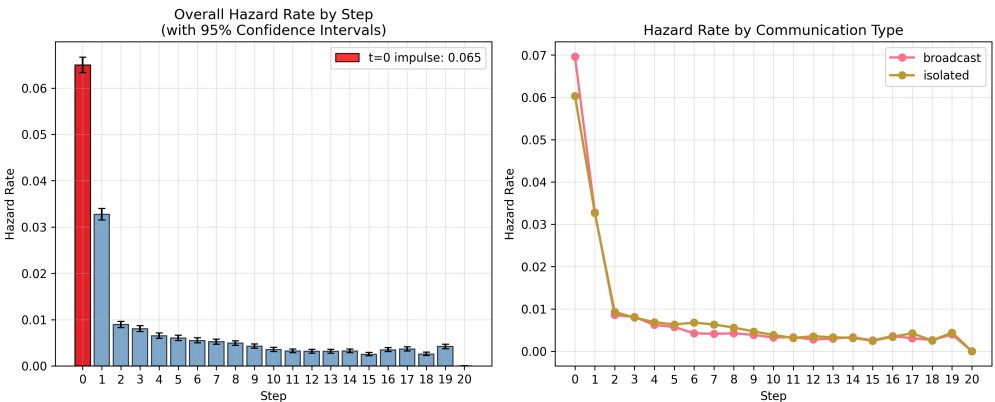

Figure 7: Discrete-time hazard by minute (pooled).

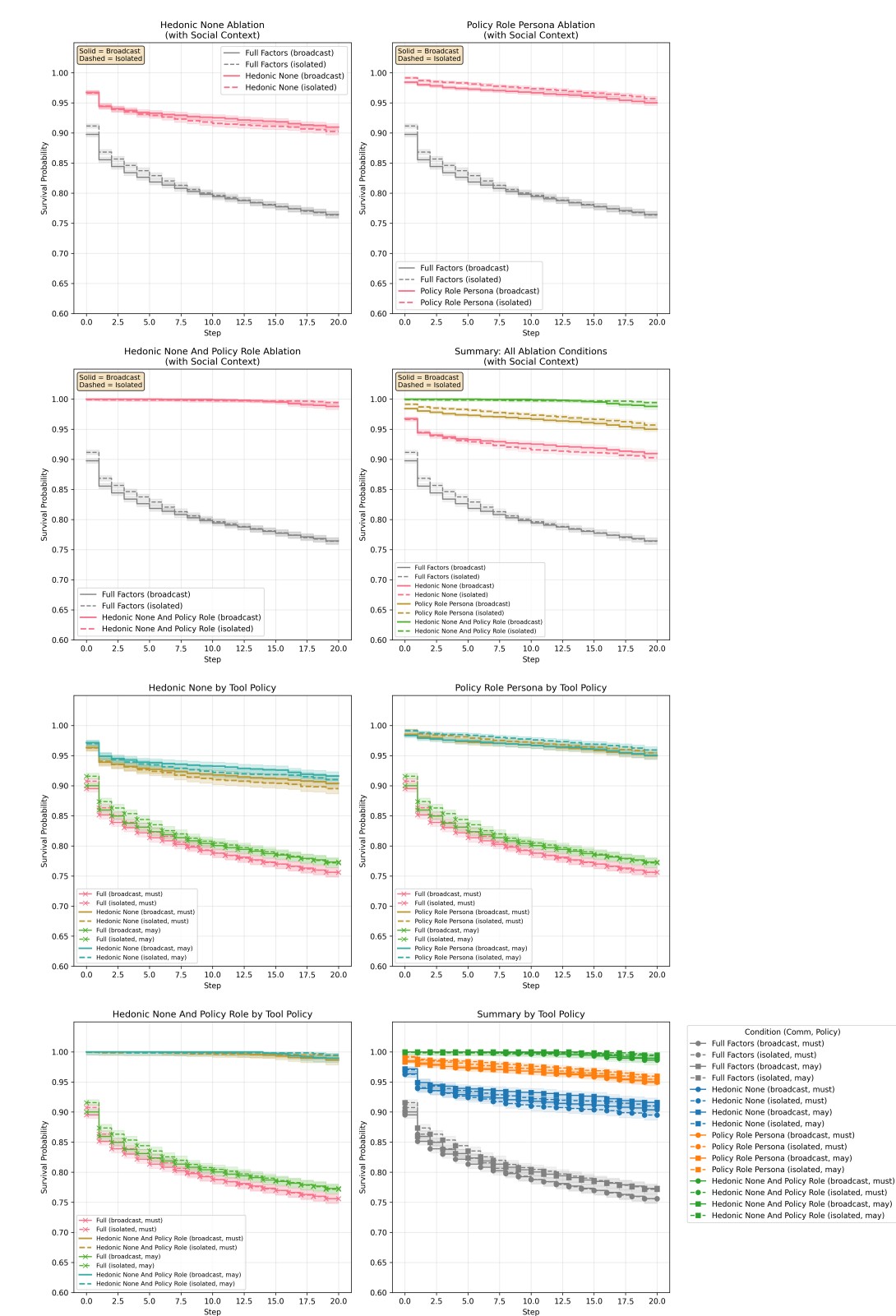

Figure 8: Kaplan–Meier survival by ablation condition. Persona removals suppress the early spike and flatten the hazard across the horizon.

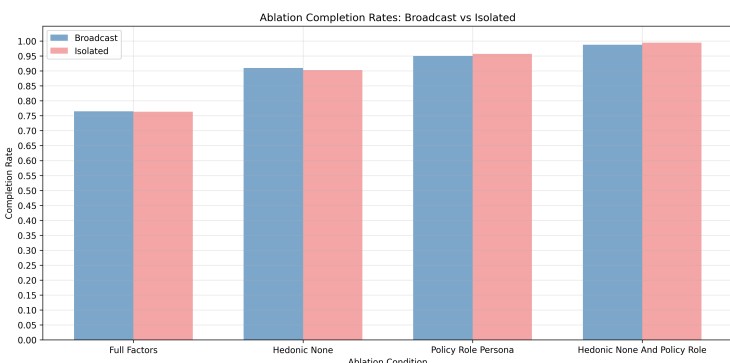

Figure 9: Completion by ablation condition and social visibility. Removing personas (hedonic, policy-role) increases completion; the combined removal approaches 1.0 and compresses the broadcast–isolated gap.

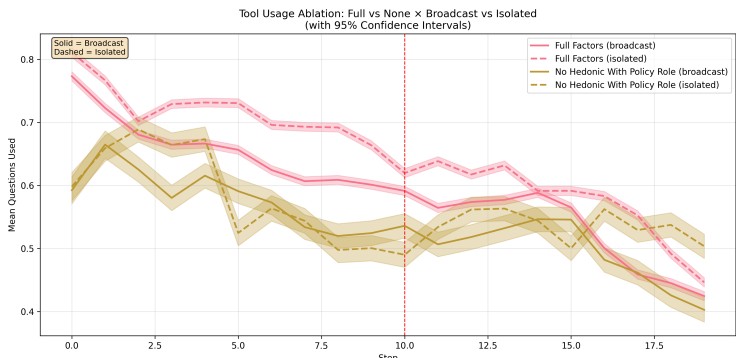

Figure 10: Tool-use under ablations: mean questions per step (with 95% CIs) for Full vs. None across social conditions. Lower question rates accompany improved survival under persona removals.

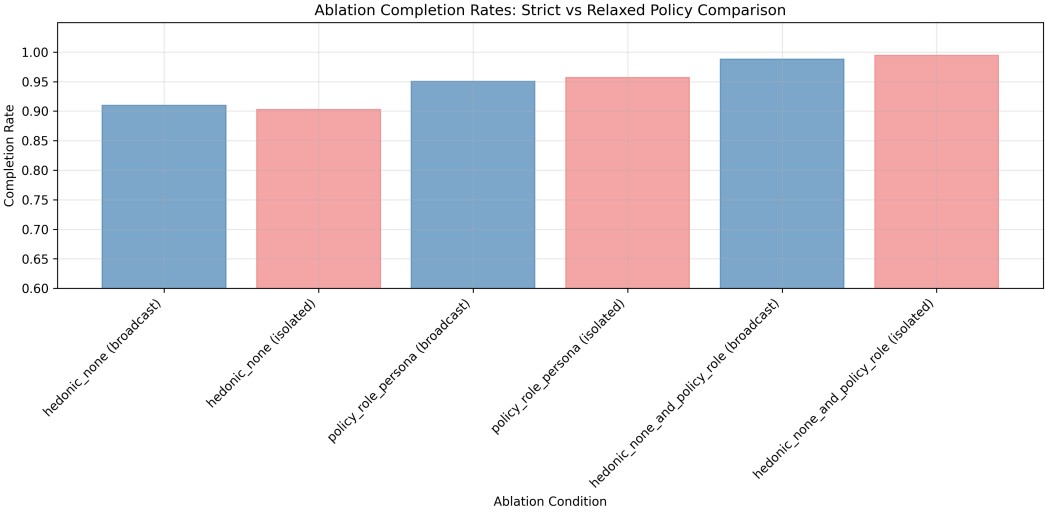

Figure 11: Alternative completion comparison (strict vs. relaxed policy view) across ablations. Results mirror the main figure: the combined removal delivers the highest completion in both social conditions.

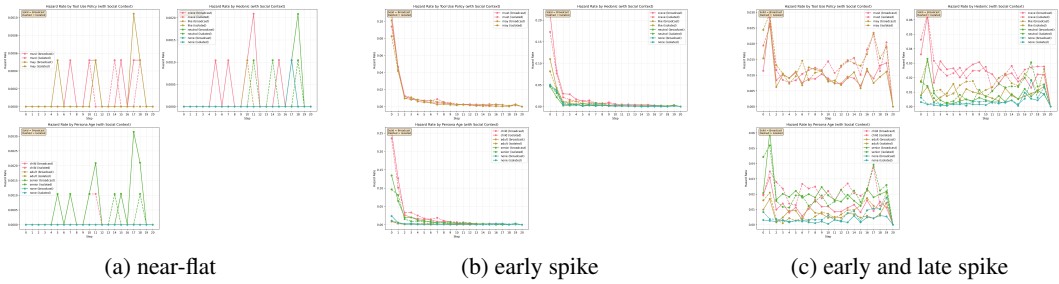

(a) near-flat      (b) early spike      (c) early and late spike

Figure 12: Identified three distinct hazard-shape regimes across the different model families, which clarify how and when social exposure influences behavior

