# OpenReview forum: "Long‑Horizon Reliability of LLM Agents: Social Exposure, Personas, and Metacognitive Policy on a Delay‑of‑Gratification Survival Benchmark"
_ICLR.cc/2026/Conference — Submitted to ICLR 2026_

### Official Review · Reviewer_MxvD · 2025-10-31

**Soundness:** 3
**Presentation:** 1
**Contribution:** 3
**Rating:** 4
**Confidence:** 3

**Summary:**

This paper introduces a microbenchmark to evaluate the long-horizonton reliability of LLM agents in multi-turn interactions. It reframes the Stanford marshmallow experiment on delayed gratification as a discrete-time survival task for LLMs under tool budget constraint. Experiments show that personas strongly affect LLMs' behavior, enforcing self-questioning increases risk of failure, and social exposure has nuanced heterogeneous effects.

**Strengths:**

1. The paper studies an interesting and meaningful problem: the behavior of LLMs in multi-turn and multi-agent interactions. The marshmallow experiment is a good case study.
2. The KM survival curve is a good way to visualize and understand the behavior of the LLM agents across time. It provides richer understanding of temporal behavior than the aggregated binary survival outcome.
3. The finding that forcing deliberation increases failure risk is a surprising result.

**Weaknesses:**

1. It is not clear to me why the authors study the persona's effect to evaluate LLMs' long-horizon reliability. The ablation studies show that removing persona pushes the completion rate to 1, making the task trivial for the LLMs. Thus, the finding that persona factors strongly modulate hazard seems to be a reflection of LLMs' ability to role-play, instead of their intrinsic long-horizon reliability.
2. The writing and presentation can be improved.
   - I feel that the presentation of the experiment results in Section 5 is quite terse, dense, and rushed.
   - Moreover, the layout of the texts and figures are quite disjointed. (For example, Table 1 and Figure 2 are mentioned in the first paragraph of Section 5, but appear in very different places.) Some key figures are deferred to the appendix (e.g., the three hazard shape regimes). It would also be better to add some visual aids for the main findings in Section 5.1.
   - I think it would help the reader's understanding if the mathematical formulation of the discrete-time hazard model, as well as the definition of the coefficient $\beta$ and the odds ratio $\mathrm{OR}$, are introduced in the main text (instead of being deferred to Appendix A.3). Currently, Section 5.1 reports results on $\beta$ and $\mathrm{OR}$ without defining them.
   - The figures' captions and legends have very small font sizes, and some figures have too many or overlapping curves, which are a bit hard to read.
   - The mathematical setup of the POMDP interaction protocol in lines 145-150 uses a small font size, which is a bit hard to read.
   - This is probably a typo: The experiments are conducted on 8 model families, but at the beginning of Section 4 (line 228), it is said that "All agents are LLM-driven using Gemini-2.5-Flash (same models across all cells and trials)."

**Questions:**

1. What is the purpose of not revealing the question budget to the LLM agents?

---

### Official Review · Reviewer_e4jz · 2025-10-31

**Soundness:** 2
**Presentation:** 1
**Contribution:** 2
**Rating:** 2
**Confidence:** 3

**Summary:**

The paper mainly discusses the reliability of LLMs in multi-turn interactions. While the authors focus on describing the simulation results and try to emphasize their contributions with various metrics, they don't offer a concise summary and thorough analysis.

**Strengths:**

The authors use LLMs to simulate multi-turn interactions and conduct discussions on multiple aspects.

**Weaknesses:**

1. The paper does not clearly outline its innovation and contributions, especially in comparison to other works on LLMs in multi-turn interactions. The paper does not provide a clear explanation of the gaps in related studies or the authors‘ motivations.

2. Parameters of the LLMs (e.g., temperature and top-p) are essential in this kind of simulation, but the authors have not included any related information or discussion.

3. The  presentation of the paper is chaotic. For instance, Figures 1A and 1B need to be redesigned to highlight the differences. The authors do not provide adequate explanations for certain terms and tools, such as what "risk horizon" (L137) and “raise a question tool” (L141) are. There is significant space left in the paper, and adding necessary explanations is important, such as for some of the terms within the ReAct framework.

4. The inconsistent and non-standard writing makes the text difficult to read. For instace, MUST and MAY are introduced in line 076, but their definitions are only provided in lines 085-086. The legends in Figure 2 and 4 need to be adjusted. Table 1 clearly needs to be improved.

**Questions:**

1. Which simulations are Figures 3 and 4 based on?

2. Why are those five hypotheses chosen?

---

### Official Review · Reviewer_Bnvk · 2025-11-01

**Soundness:** 2
**Presentation:** 1
**Contribution:** 2
**Rating:** 2
**Confidence:** 3

**Summary:**

This paper proposes a simple, long horizon multi-agent interaction benchmark modeled after the Stanford marshmellow test. The authors frame "delay of gratification" as a discrete-time survival problem and evaluate 8 LLM families with different prompt settings.

**Strengths:**

I think the adoption of survival analysis is a nice approach in the analysis. The authors have a nice factorial design. The related work section seems adequate.

**Weaknesses:**

1)	The overall writing is very dense and quite inaccessible, with lots of terminologies from both cognitive psychology and survival analysis, without providing the necessary context and thus is hard to follow. There is, for example, very little background, on what the Stanford marshmallow task is, and how exactly did the authors adopt it or why. Also there are some contradictions in the paper: Line 229 explicitly states: "All agents are LLM-driven using Gemini-2.5-Flash (same model across all cells and trials)." Yet, the abstract, Table 1, and the results section repeatedly claim that the study spans "8 model families."

2)	More importantly, the authors fail to demonstrate why is benchmark is a meaningful proxy of the broader problem of long-horizon agent reliability. The authors seek to create a “compact and auditable” benchmark but do not justify why this benchmark could measure long-horizon agent reliability. Overall, the external validity of this benchmark is not clearly shown. The author fails to justify why a simple binary-choice task in a very static environment is a good proxy for “long-horizon reliability”.

3)	As a result, it is very hard to contextualize the findings, such as those around social context or persona.

4)	A reader finishing a benchmark paper should be able to answer the question: "Who won?" or "How did the different systems compare?" This paper provides no such clarity. Instead of a summary table or leaderboard ranking the 8 model families on key metrics, the authors completely skip the main result and directly start analyzing the treatments, which is rather confusing.

5)	The paper makes strong claims about "metacognition" simply based on the mandatory use of a raise_a_question tool, which I think is unjustified, as “metacognition” is a much deeper concept.

6) Lots of formatting issues (e.g. line 48 should use inline citation; line 286 quotation is mistyped; check for duplicate citations)

**Questions:**

see above

---

### Official Review · Reviewer_Qtvf · 2025-11-03

**Soundness:** 2
**Presentation:** 3
**Contribution:** 1
**Rating:** 2
**Confidence:** 4

**Summary:**

This paper introduces a marshmallow-inspired, step-level benchmark for evaluating long-horizon reliability of LLM agents, framing delayed gratification as a discrete-time survival task and crossing three factors: social visibility (broadcast vs. isolated), persona prompts (hedonic drive; age), and metacognitive policy (mandatory vs. optional self-questioning). Using Kaplan–Meier survival, restricted mean survival time, and discrete-time hazard models over 84,540 trajectories from eight model families, the authors report a characteristic early impulse to “eat” followed by a low-hazard tail; a near-zero pooled main effect of social exposure; strong persona effects (e.g., child/senior and “crave” increase hazard, neutral/none reduce it); and a counterintuitive increase in risk under mandatory self-questioning. The study further identifies distinct hazard-shape regimes across model families and shows that ablating hedonic/age instructions flattens hazards and raises completion toward 1.0. The authors release prompts, logs, and analysis artifacts to support full replication and future work on social exposure and tool-policy design in multi-turn agent settings.

**Strengths:**

1. The paper presents an perspective connecting LLM behavioral dynamics with human cognitive psychology, particularly through the “Marshmallow” experiment.
1. The writing is clear, logically structured, and easy to follow.

**Weaknesses:**

1. The paper’s technical contribution is limited. It primarily reinterprets behavioral evaluation rather than introducing new techniques, insights, or agent architectures. As a result, its relevance and impact within the LLM agent research domain are unclear.
1. The reported findings (H1–H5) mainly describe statistical outcomes without deeper interpretation or discussion of  implications of insights for LLM agents.
1. The results shown in Figure 2 appear highly model-dependent, raising concerns about generalizability and robustness across models.
1. The reward design is overly simplistic; the value of “eating the marshmallow” varies by persona, for example, an adult with diabetes may find consumption undesirable.

**Questions:**

1. Why does Mean Peers Eaten So Far decrease in Figure4(b)? Can a peer transfer from eaten to not eaten?
1. How many peers under social exposure scenario? Are they all the same to the peer in the isolated scenario?

---

### Meta-Review · Area_Chair_MxpY · 2026-01-08

**Summary:**

The reviewers raised several main concerns:
1. Limited contribution with unclear contextualized relevance and impact (Qtvf, Bnvk, e4jz, MxvD)
2. Presentation is difficult to follow (Bnvk, e4jz, MxvD)
3. Limited interpretation and analysis of results (Qtvf, Bnvk)
4. Results appear to be highly model-dependent (Qtvf)

**Reviewer Concerns:**

Unfortunately the authors did not submit a response during the rebuttal phase. Therefore, these concerns remain outstanding.

**Reviewer Scores:**

Due to the lack of rebuttal, all reviewers are virtually certain to keep their original ratings.

---

### Decision · Program_Chairs · 2026-01-26

Reject